**∂ | Open Peer Review** | Conduct of Scientific Research | Commentary

# Fiber, microbiomes, and SCFAs: insights from companion animal models to inform personalized nutrition

Leigh A. Frame[1,2,3,4]

**ABSTRACT** A recent study by A. Bhosle, M. I. Jackson, A. M. Walsh, E. A. Franzosa, et al. (mSystems 10:e00452-24, 2024, https://doi.org/10.1128/msystems.00452-24) enhances our understanding of dietary fiber's impact on the gut microbiome and metabolome in companion animals, uncovering individual variations in microbial and metabolic responses. By examining short-chain fatty acid (SCFA) profiles in response to fiber, the authors reveal potential therapeutic benefits of tailored dietary interventions, such as enhanced gut and immune health. These findings resonate with human microbiome research, where dietary fiber has shown health benefits through microbial diversity and SCFA production. The study emphasizes the potential for breed-specific responses to fiber, given the variation in microbiome composition and physiology across breeds. Such insights align with emerging concepts of personalized nutrition, offering an opportunity to develop precision dietary strategies that address specific health needs in both veterinary and human contexts. This foundational research positions dietary fiber as a valuable tool in preventive health, providing a roadmap for future studies to refine individualized approaches for gut microbiome modulation.

**KEYWORDS** dietary fiber, gut microbiome, metabolome, companion animals, personalized nutrition, microbiome diversity, veterinary nutrition, microbiome-diet interactions, short-chain fatty acids, translational research

A recent study by Bhosle et al. (1) contributes significantly to understanding how dietary fibers influence gut health, focusing on individualized microbiome responses. Their design highlights a growing paradigm in microbiome research: the complex interplay between diet, genetics, and the microbiome that shapes individual dietary responses (Fig. 1). This concept of personalized nutrition, observed in human studies, may pave the way for tailored strategies in veterinary and human care.

## TRANSLATIONAL INSIGHTS AND COMPANION ANIMAL MODELS

Bhosle et al. (1) emphasize the advantage of studying food-microbiome interactions in companion animals that share human-like environments but have more consistent, controlled diets. These animals provide a relevant model for understanding food-microbiome relationships, which are challenging to isolate in human studies, i.e., dietary variability (2). The authors find that diet explains more microbiome variation in this dog cohort (≈16%) than in free-living humans (<1%–10%), likely due to controlled diets, though less so than in laboratory mice (≈35%). Such findings recapitulate the functional robustness seen in humans, where key functions such as energy harvest and metabolite production are conserved by the ecosystem, though not necessarily individual microbes, despite dietary shifts, further supporting model validity.

Address correspondence to Leigh A. Frame, leighframe@gwu.edu.

The author declares no conflict of interest.

*The views expressed in this article do not necessarily reflect the views of the journal or of ASM.*

See the original article at https://doi.org/10.1128/msystems.00452-24.

**FIG 1** Conceptual framework of a growing paradigm in microbiome research: the complex interplay between diet, genetics, and the microbiome that shapes individual dietary responses.

## INDIVIDUALIZED RESPONSE TO DIETARY FIBER

In both veterinary and human fields, dietary fiber profoundly impacts the gut microbiome, particularly through microbial production of short-chain fatty acids (SCFAs) like butyrate, acetate, and propionate. Bhosle et al. (1) demonstrate that different fibers in canine diets produce distinct SCFA profiles, especially butyrate and propionate. This parallels human research, where fiber has shown benefits via SCFA production, supporting gut health, immune resilience, and neuroprotective effects (3). The authors underscore the variability in microbiome composition across individual dogs, mirroring findings in human studies where differences in microbiota composition influence dietary responses (2). As prior research shows, factors like diet, genetics, and environmental exposures shape the microbiome, suggesting that companion animals may benefit from personalized dietary strategies that account for these variables.

## SCFA PRODUCTION

The specific SCFAs produced by gut microbiota are critical metabolites with effects extending beyond the gut. SCFAs, particularly butyrate, support gut barrier integrity, a key factor in maintaining immune homeostasis. The study's focus on butyrate aligns with human research linking butyrate production to improved inflammation and gut health and potential neuroprotective effects. Emerging research suggests that SCFAs can influence epigenetic mechanisms, such as histone deacetylase inhibition, impacting immune responses (3). This finding opens possibilities for dietary interventions targeting SCFA profiles to support animals with specific needs or conditions—an area in need of better tools.

## BREED-SPECIFIC INTERVENTIONS

Bhosle et al. (1) suggest the need for studies examining breed-specific responses. Differences in microbiome composition and dietary response could be due to breed-specific traits, such as variations in digestive anatomy. For instance, carnivores have shorter digestive tracts with quicker transit times and, therefore, likely lower production of endogenous SCFAs. While this may seem counterfactual to the idea of the centrality

of SCFAs in all species, it may, instead, be a marker of gut length, as longer guts may require greater SCFAs for maintenance (gut barrier integrity, as discussed under "SCFA production" above).

In humans, genetic and microbiome composition influence fiber fermentation rates and SCFA production, indicating that certain fibers may benefit specific microbiomes/breeds more than others (2, 4, 5). Tailoring fiber sources to breed-specific microbiome characteristics may enhance health outcomes, especially for breeds prone to gastrointestinal or metabolic issues. As they note, the authors cannot draw firm conclusions with a limited sample size of mixed breed dogs ($n = 3$) versus a single pure breed ($n = 15$ beagles), all similarly sized.

## MICROBIOME DIVERSITY AS A MARKER OF HEALTH

A consistent finding in gut microbiome research is that a diverse microbiome generally correlates with better health outcomes (6–8). Bhosle et al. (1) show how different fibers promote varying levels of microbial diversity in the canine gut. In humans, diets rich in diverse plant fibers are associated with a more diverse microbiome, supporting resilience against infections, inflammation, and chronic diseases (2, 3). This study reinforces that dietary interventions promoting microbial diversity may be a key preventive approach in both humans and companion animals.

## TRANSLATIONAL INSIGHTS AND FUTURE DIRECTIONS

Bhosle et al. (1) emphasize the unique value of companion animals as models for studying food-microbiome interactions. Their controlled yet human-like environments provide a relevant framework for understanding dietary impacts. This study builds upon previous work by the same group—comprehensive gut microbiome profiling of companion animals (2,272 dogs and 367 cats); this foundational research identified key species-level genome bins unique to companion animals, advancing our understanding of host-specific microbiome adaptations (9). Together, these findings enhance the broader translational value of companion animal models for exploring dietary interventions with potential applications in both veterinary and human health.

It is clear that the findings of Bhosle et al. (1) have implications beyond veterinary nutrition, contributing to emerging research on the gut microbiome as a mediator between diet and health outcomes and offering insights applicable to both animal and human well-being. The growing recognition of SCFAs' impact on immune function, metabolism, and neuroinflammatory pathways suggests that dietary fiber interventions may have far-reaching benefits. Given the connections between the gut microbiome, immune responses, and even behavior, as seen in neuroinflammation research, future research on such dietary interventions is warranted to assess cross-species benefits in managing disease and promoting well-being.

Future research may benefit from longer feeding periods and washout phases to assess more robust, sustained impacts of fiber on the gut microbiome. While randomized design is essential in human studies, given the feasibility and the complexity of microbiome research, a more balanced approach may be appropriate in companion animals. Given the intricate relationships between the diet and the gut microbiome, longitudinal studies would enable researchers to clarify the long-term impacts of fiber intake on this complex ecosystem. Such studies could provide further evidence supporting dietary fiber as a potential therapeutic for a range of health concerns, from gastrointestinal to metabolic and immune-related issues.

## CONCLUSIONS

This study is a valuable addition to the field, underscoring the importance of dietary fiber in shaping the gut microbiota and metabolome in companion animals. In congruence with human studies, Bhosle et al. (1) advance personalized dietary strategies, laying the foundation for breed-specific exploration and demonstrating how dietary fiber supports

well-being across species. With varied responses to fiber types, dietary diversity remains essential in nutrition counseling for humans and is likely the case for companion animals as well.

By emphasizing individualized responses and specific SCFA profiles, this study accentuates the need to view the microbiome as a complex, adaptive system influenced by diet, genetics, physiology, etc. Bhosle et al.(1) take an important step toward integrating microbiome-based dietary strategies into veterinary care, supporting precision nutrition with transformative potential for animal and human well-being.

## AUTHOR AFFILIATIONS

[1]The Frame-Corr Laboratory, Department of Clinical Research and Leadership, The George Washington University School of Medicine and Health Sciences, Washington, DC, USA
[2]Resiliency & Well-being Center, The George Washington University School of Medicine and Health Sciences, Washington, DC, USA
[3]Office of Integrative Medicine and Health, The George Washington University School of Medicine and Health Sciences, Washington, DC, USA
[4]Department of Physician Assistant Studies, The George Washington University School of Medicine and Health Sciences, Washington, DC, USA

## AUTHOR ORCIDs

Leigh A. Frame http://orcid.org/0000-0002-1475-2778

## AUTHOR CONTRIBUTIONS

Leigh A. Frame, Conceptualization, Writing – original draft, Writing – review and editing

## ADDITIONAL FILES

The following material is available online.

Open Peer Review

**PEER REVIEW HISTORY (review-history.pdf).** An accounting of the reviewer comments and feedback.

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

## AUTHOR BIO

**Leigh A. Frame**, PhD, MHS, is an associate professor at The George Washington University School of Medicine and Health Sciences, where she serves as executive director of the Office of Integrative Medicine and Health (OIMH) and associate sirector of the GW Resiliency & Well-being Center. At OIMH, Dr. Frame leads initiatives in education, research, and community outreach promoting whole-person care. At the Resiliency & Well-being Center, she oversees its four-pillar mission: education, clinical care, communications, and research/metrics. Dr. Frame's research experience includes roles at the Johns Hopkins Center for Bariatric Surgery and the National Institutes of Health National Institute of Neurological Disorders and Stroke Parkinson's Disease Biomarker Discovery Program. She is building a translational research program including the microbiome-gut-brain axis, nutritional immunology, and integrative health. Dr. Frame holds a PhD in human nutrition and an MHS in molecular microbiology and immunology from the Johns Hopkins Bloomberg School of Public Health.

