## [Reviewer comments · mSystems]

Commentary on “Response of the Gut Microbiome and Metabolome to Dietary Fiber in Healthy Dogs”

Leigh Frame

Corresponding Author(s): Leigh Frame, The George Washington University School of Medicine and Health Sciences

Review Timeline:

Submission Date:	November 19, 2024
Editorial Decision:	December 17, 2024
Revision Received:	January 7, 2025
Accepted:	January 11, 2025

Editor: Katrine Whiteson

Reviewer(s): The reviewers have opted to remain anonymous.

Transaction Report:

DOI: <https://doi.org/10.1128/msystems.01454-24>

Re: mSystems01454-24 (Commentary on "Response of the Gut Microbiome and Metabolome to Dietary Fiber in Healthy Dogs")

Dear Dr. Leigh A. Frame:

Thank you for the privilege of reviewing your work. Below you will find review comments and instructions from the mSystems editorial office. We request minor modifications to improve an already strong perspective, and to aid the process to give an opportunity to submit the necessary forms (see details below).

Revision Guidelines

Sincerely,
Katrine Whiteson
Senior Editor
mSystems

Overall, this is a really nice perspective, even with a wonderful schematic figure, and very generous of the authors to invest time in this. The following comments/suggestions are meant to strengthen an already nicely written piece.

Comments:

1. One general comment that would be helpful to address is what the ideal fiber intake for a dog is, and how this compares to humans and other companion animals. Does increasing fiber intake make sense for dogs? Are dogs more like omnivores or

carnivores, and does their gut design reflect this?

2. Line 68: Consider specifying that this refers to the gut microbiome, as higher diversity in the gut microbiome is associated with health, but this is not true for oral, vaginal and other systems

3. Line 61-63, and 75: I appreciate comments related to digestive anatomy. For findings beyond veterinary medicine, I think it would be worth framing this in terms of ideal fiber intake depending on evolutionary history and gut anatomy and physiology. Do more carnivorous animals still depend on SCFAs? They have lower gut microbial loads, I think... My own reading took me to Inuit diets, and I was surprised to learn that Inuit populations in Northern Canada had similar fiber intake as urban people from Toronto: Dubois, G., Girard, C., Lapointe, FJ. et al. The Inuit gut microbiome is dynamic over time and shaped by traditional foods. *Microbiome* 5, 151 (2017). <https://doi.org/10.1186/s40168-017-0370-7>

4. The Huttenhower group also just published a more comprehensive analysis of companion animal microbiomes with a nice inclusion of the unknowns (~75% unknowns in the dog microbiomes), and it might make sense to cite that one here: Branck T, Hu Z, Nickols WA, Walsh AM, Bhosle A, Short MI, Nearing JT, Asnicar F, McIver LJ, Maharjan S, Rahnavard A, Louyakis AS, Badri DV, Brockel C, Thompson KN, Huttenhower C. Comprehensive profile of the companion animal gut microbiome integrating reference-based and reference-free methods. *ISME J.* 2024 Jan 8;18(1):wrae201. doi: 10.1093/ismejo/wrae201. PMID: 39394961; PMCID: PMC11523182.

@@@ Please complete the additional submission form questions only available Resubmission stage before acceptance as described above

Response to Reviewer

1. One general comment that would be helpful to address is what the ideal fiber intake for a dog is, and how this compares to humans and other companion animals. Does increasing fiber intake make sense for dogs? Are dogs more like omnivores or carnivores, and does their gut design reflect this?

Thank you for your review. As I am a human nutrition microbiome expert and not a canine expert, I did not feel comfortable speaking to the state of the science for canines. Instead, I focused on the quality of the science and putting this study in context in the nutrition and microbiome field broadly. This information would be more pertinent in the article itself, rather than the commentary.

2. Line 68: Consider specifying that this refers to the gut microbiome, as higher diversity in the gut microbiome is associated with health, but this is not true for oral, vaginal and other systems

Excellent point. Corrected. Thank you for catching that!

3. Line 61-63, and 75: I appreciate comments related to digestive anatomy. For findings beyond veterinary medicine, I think it would be worth framing this in terms of ideal fiber intake depending on evolutionary history and gut anatomy and physiology. Do more carnivorous animals still depend on SCFAs? They have lower gut microbial loads, I think... My own reading took me to Inuit diets, and I was surprised to learn that Inuit populations in Northern Canada had similar fiber intake as urban people from Toronto: Dubois, G., Girard, C., Lapointe, FJ. et al. The Inuit gut microbiome is dynamic over time and shaped by traditional foods. *Microbiome* 5, 151 (2017). <https://doi.org/10.1186/s40168-017-0370-7>

I am not sure this is purely germane to my commentary, but I have added a statement to expand on my point in the hope of clarifying this.

4. The Huttenhower group also just published a more comprehensive analysis of companion animal microbiomes with a nice inclusion of the unknowns (~75% unknowns in the dog microbiomes), and it might make sense to cite that one here: Branck T, Hu Z, Nickols WA, Walsh AM, Bhosle A, Short MI, Nearing JT, Asnicar F, McIver LJ, Maharjan S, Rahnavard A, Louyakis AS, Badri DV, Brockel C, Thompson KN, Huttenhower C. Comprehensive profile of the companion animal gut microbiome integrating reference-based and reference-free methods. *ISME J.* 2024 Jan 8;18(1):wrae201. doi: 10.1093/ismejo/wrae201. PMID: 39394961; PMCID: PMC11523182.

Thank you for the recommendation. I have added this information to the *Translational Insight and Future Directions* section.

Re: mSystems01454-24R1 (Commentary on "Response of the Gut Microbiome and Metabolome to Dietary Fiber in Healthy Dogs")

Dear Dr. Leigh A. Frame:

Your manuscript has been accepted, and I am forwarding it to the ASM production staff for publication. Your paper will first be checked to make sure all elements meet the technical requirements. ASM staff will contact you if anything needs to be revised before copyediting and production can begin. Otherwise, you will be notified when your proofs are ready to be viewed.

Sincerely,
Katrine Whiteson
Editor
mSystems